# Patients' experience of suffering a distal radius fracture with long-lasting impairment —A qualitative study

**Hanna Südow** [1,2]*, **Cecilia Mellstrand Navarro**[3], **Sari Ponzer**[1], **Caroline Olsson**[4]

**1** Department of Clinical Science and Education, Södersjukhuset, Karolinska Institutet, Stockholm, Sweden,
**2** Department of Orthopedic Surgery, Södersjukhuset, Karolinska Institutet, Stockholm, Sweden,
**3** Department of Clinical Science and Education, Danderyds Sjukhus, Karolinska Institutet, Stockholm, Sweden, **4** Department of Environmental Medicine, Karolinska Institutet, Stockholm, Sweden

☯ These authors contributed equally to this work.
* hanna.sudow@ki.se

**Data Availability Statement:** The datasets generated and analysed during the current study are not publicly available due to the Swedish ethical review regulation. Data are available upon reasonable request. Inquiries for data access

## Abstract

### Background

Every year a large number of people suffer a distal radius fracture and some of them never regain their activity level. The correlation between radiographic features and outcome explains some but not all the disability perceived after fracture healing and rehabilitation. Little is known of the persons reporting persistent upper limb dysfunction. The aim of this study was to improve the understanding of the experience of persistent impairment, treatment, and recovery process after a distal radius fracture, with a focus on patients with benign radiographic and injury features but without full restoration of function.

### Materials and methods

This is a qualitative study performed through semi structured interviews analyzed with content analysis. The participants had previously been treated at Södersjukhuset hospital, Stockholm, Sweden for a distal radius fracture and reported that they had not regained their previous function. The sampling was purposeful and participants who were assumed to carry a lot of information were selected to participate. All 17 interviews were recorded, transcribed, and coded. Codes were grouped and categories formed.

### Results

Three main categories were inductively identified from the data. 1. Limitations in life due to persistent impairment—the description of what was not regained, physically, mentally, and activity-wise. 2. Being a patient—the participants' description of experiences and feelings when assuming the role as a patient. 3. The last main category identified was personal circumstances such as having to care for children, being alone or having a partner, having a demanding employer or the person's inherent personality and attitude.

should be sent to Institutionen för Klinisk forskning och utbildning Södersjukhuset, Karolinska Institutet, Sjukhusbacken 10, 118 83 Stockholm or contact the corresponding author, Hanna Südow, hanna.sudow@ki.se, who will then contact the Swedish Ethical Review Authority for permission to share the data.

**Funding:** Open access funding provided by Karolinska Institute. This study was funded by the Swedish Research Council and the Regional Agreement on Medical Training and Clinical Research between the Stockholm County Council and Karolinska Institute (ALF). CMN was supported by Region Stockholm (clinical research appointment) The funders had no role in study design, data collection and analysis, decision to publish, or preparation of the manuscript.

**Competing interests:** The authors have declared that no competing interests exist.

## Conclusion

Patients suffer from impairment difficult to measure after a distal radius fracture. During the treatment and recovery process their experience as being patients, their perceived level of knowledge and their personal circumstances all play important roles in understanding how the participants experienced their recovery.

## Introduction

Every year a large number of people sustain a distal radius fracture. It is the most commonly treated fracture in an overall population [1, 2] causing a substantial suffering. The incidence rate is reported to 15-30/10 000 person years [1–7].

The incidence rate is highest in postmenopausal women [2, 3, 8] due to a high degree of osteoporosis [9, 10]. With an aging population the number of distal radius fractures can be assumed to increase. Further knowledge in the area is needed to minimize fracture sequelae and provide cost effective healthcare.

The severity of distal radius fractures varies from very benign fractures to complex and severe injuries and many affected persons will suffer long term pain and loss of function [11–13]. Immediately after the injury nearly everyone is limited in their daily activities. Within three months pain and loss of function are usually highly improved. However, by a year as many as 8% have been reported to experience moderate to severe impact in their activities [14]. Patients with a slow recovery after a distal radius fracture have been shown to have a higher rate of depression [15] but temporality and causes have not been determined.

An impaired wrist function, and thereby an impaired hand function, affects most aspects of daily life and working ability. Any heavy lifting could be impossible to perform without pain, as well as different household chores and other daily activities such as getting dressed, turning a key or pour coffee from a pot. The disability may lead to loss of independence, loss of income and affect relationships [16, 17] The correlation between the radiological and clinical outcome has been extensively investigated [9, 11, 18] but other factors that could predispose an impaired recovery after a distal radius fracture remain unknown. To clarify other explanations than radiologic features affecting outcome, qualitative studies may yield new knowledge. Andreasen et al [16] published a report of patients scheduled for corrective osteotomy. Their findings indicated that living with a malunited distal radius fracture led to "obstacles in every-day life" and caused major struggles. All patients in Andreasen's study had fractures healed in malunion and thus they had a clear radiological explanation for their impairment. Clinical experience indicate that many patients suffer from extensive disability even when the objective radiological and physiological results seem good. There are no studies published investigating patients with an experienced bad outcome although radiography findings are close to normal.

The aim of this study was to improve the understanding of patients' experiences of not feeling fully recovered after treatment of a distal radius fracture, with a focus on patients with benign radiographic and injury features, thereby shedding light on non-radiographic factors predisposing a poor outcome.

## Methods

### Study design

This is a qualitative interview study using content analysis. Qualitative methods are well suited [19] when there is little or no previous knowledge about a phenomenon. It is also suitable to

gain deeper knowledge about people's experiences or thoughts. The design was chosen to add depth to the knowledge of the patients' experience of a poor recovery after a distal radius fracture in cases with an expected good or excellent prognosis.

## Setting

The study was conducted at the Orthopedic department at Södersjukhuset in Stockholm, Sweden. The Södersjukhuset hospital is a second level trauma center with a catchment area of approximately 675 000 inhabitants [20]. In Sweden all trauma health care is publicly funded and freely available to all citizens. Employed and non-employed persons who are injured or ill have the right to a paid sick leave if the loss of function impairs activities needed for their professional or work-seeking duties. In Stockholm, patients with a fracture to the wrist are treated either at a primary care emergency unit at a hospital or at a separate emergency unit outside the hospitals, and then referred to an orthopedic clinic for further treatment. When presenting to the emergency unit with a distal radius fracture, X-ray is used to confirm the diagnosis and the fracture is reduced when necessary. A plaster cast is nearly always applied, and the patient will either undergo surgery within one to two weeks or be evaluated after about 10 days to determine if secondary surgery is needed or not [21]. The patient is then referred to an occupational therapist either at the hospital (surgically treated patients) or in the primary care.

## Sampling

A purposeful sampling was chosen in order to recruit study participants with experiences of sustaining a distal radius fracture and not regaining function. Both surgically treated and non-surgically treated patients were included and both genders were represented but without any aim of an even distribution. Study participants were recruited from rehab units where the occupational therapists identified individuals with a poor outcome (n = 7), from the surgical waiting list for plate removal surgery (n = 1) and found through the Swedish fracture register (n = 1). Two participants were identified by Orthopedic surgeons at the unit who knew about suitable patients for the study (n = 2). Other individuals were found by sending information about the study to former patients and asking them to participate if they did not feel recovered (n = 6).

When a potential study participant was identified, a brief review of radiographs and medical records was made to evaluate if they were eligible. Patients with obvious reasons for a poor outcome such as open fractures, additional fractures at the same time, previous rheumatism, substance abuse or dementia were excluded. The recruitment process started in May 9th 2022 and was completed May 29th 2023.

Written information of the study was sent, and a signed consent form was returned in cases they agreed to participate. Further oral information was provided by the time data collection started.

## Data collection

All interviews took place at the Orthopedic outpatient clinic between October 2022 to May 2023 and were performed by a female resident in Orthopedic surgery (HS) who had not taken any role in the previous treatment. Civil clothes were worn to minimize the doctor-patient relationship during interview, and investigations were conducted in a neutral office. A semi-structured interview guide constructed by the research group was used to capture different aspects of life with and in the aftermaths of a distal radius fracture. After the first two interviews slight adjustments to the interview guide were made.

All interviews started with an open-ended question only encouraging the study participants to talk freely about their experience.

Probing was utilized as needed to deepen the story. More specific questions were asked regarding pain, function, limitation, work, sick leave, Covid-19, and dominant hand as needed. All interviews were, with the study participants' consent, recorded and transcribed verbatim and notes were made during and after the interviews. After 17 interviews the research group found the data to be sufficient to answer the research questions.

## Data analysis

Inductive content analysis as described by Elo and Kyngäs [22] was conducted. Only manifest content was analyzed, meaning that the analysis stayed close to what the study participants said. After listening to the recordings and going through the transcripts and becoming immersed in the data, meaning-bearing units were identified, consolidated, and coded by the first author (HS) using NVivo17.1 for Windows. The codes were data derived since an inductive approach was used. After this open coding phase, codes were grouped by HS and discussed in depth multiple time with the other authors. Codes, as well as main-, sub- and generic categories, were discussed until consensus was reached.

## Trustworthiness

The study participants were chosen to illustrate different aspects of the experience under investigation. Participants were strategically sampled through multiple sources with the purpose to broaden the data. The constitution of the research group a resident in orthopedic surgery and PhD student at the time (HS), a hand surgeon and associate professor (CMN) with expertise in distal radius fracture, and a senior Orthopedic surgeon and professor with extensive research experience (SP) and a sociologist and PhD with expertise in qualitative research (CO), provided different aspects to the planning and analysis to increase the trustworthiness.

## Ethics

This study was performed according to the World Medical Association's Declaration of Helsinki [23] and approved by The Swedish Ethical Review Authority (2020–01919).

## Results

After 17 participants were interviewed, the research group agreed the data was rich enough to illustrate the research question. Two participants were males and 15 were females. Ages spanned between 33 and 81 years (see S1 Appendix for more detailed demographics including patient reported outcome measures). All participants consented to being tape-recorded with a total recorded time of 633 minutes giving a mean of 37 minutes ranging from 15 to 59 minutes.

The analysis resulted in three main categories: Persistent impairment and how it affects life, Being a patient and Personal circumstances affecting the experience. The main categories were derived from generic and sub-categories (Table 1).

### Persistent impairment and how it affects life

There was a constant awareness of the hand and wrist, reported in all participants with a varied degree of limitation in their life. Activities could be performed with extra time or struggle or avoided all together and it was not only impairment in the wrist that led to limitations.

**Table 1. The constructed categories in a qualitative study of the experience of a distal radius fracture.**

| Subcategory | Generic category | Main Category |
|---|---|---|
| Personal care and house-hold chores, i.e. cooking, cleaning, dressing | Being limited in activities | Persistent impairment and how it affects life |
| Work related activities | | |
| Limitation in sport activities | | |
| The perception of how strength and movement are restored | Perceived function, pain and complications in the hand or the rest of the body | |
| Pain that still bothers the participants | | |
| Reduced dexterity | | |
| An unfamiliar connection and awareness of the hand | | |
| Anxiety, sadness, guilt, and fear | Effects on the mind and general wellbeing | |
| Stress | | |
| Lack of energy | | |
| The feeling of not being in control, feeling alone and vulnerable | Having to trust and rely on others | Being a patient |
| Having to trust healthcare providers and not knowing their experience level | | |
| Difficulties getting in contact with the health care provider | Navigating the health care organization and experiencing its limitations | |
| Wait for treatment and evaluation | | |
| Experience surgery or other procedures for the first time | | |
| Facts that were considered necessary to feel comfortable | Desire for knowledge | |
| How the delivery of information affected what was learned | | |
| Learning from other patients | | |
| Dependency on someone else or others depending on them, i.e. children, partner etc. Physical activities. Economical resources | Life circumstances | Personal circumstances affecting the experience |
| Mentally or physically challenging work | Work related circumstances | |
| Opportunity to adjustments of work environment or chores. | | |
| Taking the responsibility or put it on others | Reflection of personality attributes affecting the experience | |
| Attitudes | | |

**Being limited in activities.** A wide range of daily activities were affected. Participants had issues with personal care like washing their hair, using lotion, getting dressed or brushing teeth and had strategies to overcome or avoid the issues. "You can't hold the toothbrush with enough strength needed to brush and you can't do that movement, that circular movement. It's impossible. For a long time, I had to use my left hand and got an electrical toothbrush." (PN 006) Household chores led to difficulties with cleaning, washing, and carrying shopping bags. Unlocking or opening doors was described to be a major struggle to perform onehanded. Cooking was difficult due to stiffness causing problems cutting and peeling etc. but also due to inability in lifting heavy objects such as a frying pan or pasta pot. A feeling of sensitivity to high and low temperatures was described to make cooking even less feasible. Solutions reported included adapting what to eat and using semi-finished products.

The impairment impacted their life, body and mind and was described to change the self-perception and outlook on the future. Some tasks at work like handwriting or using a drill in the affected hand still issued some difficulties for the participants.

**Perceived function, pain and complications in the hand or the rest of the body.** The wrist and hand didn't feel like before the injury. Both the inability to move the wrist as previously but also a feeling of resistance in the movement and sometimes with a persistent pain that was still bothering. Limitation in all ranges of movements was described with an emphasis on pronation, supination and finger mobility or combined motions with or without weight load. Inability to leaning with the hands on a table did not limit any activity but still acted as a constant reminder of the impairment. Furthermore, a lack of dexterity was noticed by participants. There were descriptions of the hand and wrist feeling rigid after inactivity. Carrying

anything heavy was difficult or even impossible due to pain or loss of strength and sometimes due to fear of the notion that the activity would be harmful. The persistent pain could be intense and sometimes led to avoidance of activities to minimize the suffering.

Participants described the hand as not completely belonging to the body, a vague elusive feeling that was almost always present and difficult to put into words. Even when everything else was fine this feeling was persistent. "It feels like the hand is screwed into place, or like a hans on a LEGO-man–clamped there." (PN 008). "I don't really trust it because it doesn't feel like a natural part of my body yet" (PN 004). "It is like the signals don't. . . don't really go through yet." (PN 002). "I don't really feel limited any longer, but I feel my wrist all the time." (PN 016) There were also descriptions of a band like discomfort that was bothering more than painful. "It is just a feeling that someone holds a hand around my wrist, or a tape" (PN 017)

**Effects on the mind and general wellbeing.** To be injured was described as a trauma and it led to fear of falling again or caused anger towards the society or towards persons who caused the accident. It also led to a feeling of fragility and an awareness of being mortal causing fear, anxiety, and sadness.

A general exhaustion was described leading to inability to handle other aspects in life. Some described that life had slowed down with a consequent inability to catch up. "It feels like I am behind with everything somehow, I still haven't been able to catch up with anything. . . all projects I had started in my home just. . ." (PN 012).

## Being a patient

Most individuals are not used to being a patient. Regardless of if it was a new experience or not, the encounter with the health care system yielded emotions of being forced into the role as a patient. Trusting and needing to be cared for was intimidating and unfamiliar and a thirst for knowledge was often not satisfied.

**Being in the hands of others and having to trust.** In the unfamiliar situation and with pain and uncertainty, the feeling of being vulnerable, small, and lonely was striking. It was described to be important to feel seen and listened to and if that wasn't fulfilled it led to an experience of loneliness, insecurity, anxiety, and distrust of the healthcare system. Some participants doubted the competence of the health care personnel, feeling like victims of medical malpractice. Lack of or inaccurate information was described to make it very stressful to be in the hands of others.

Positive descriptions of awareness of the human connection, observations of health care workers taking their time and making an extra effort led to a sentiment of being in good hands, feeling safe and prioritized.

When adequate and understandable information was provided, the participant felt safe and content with the orthopedic surgeon responsible and their choice of treatment. "I felt really safe with my doctor. She told me already from the beginning it was a severe fracture and told me about the plan and that I would need surgery. . . . . .I feel that it can't be a joint decision– she is the doctor and has the knowledge. I didn't." (PN 9) Without information or when forced to make the decision of treatment themselves the situation allowed a feeling of insecurity.

Reassurance and encouragement felt important and reduced the anxiety and fear of making a mistake. Adequate information and confirmation provided a satisfying feeling of being able to strive forwards and getting back in charge.

**Navigating the health care organization and experiencing its limitations.** Understanding and handling the wait for surgery was by some participants expressed as concerns about staff shortage, and fear that loss of production during holidays would affect the waiting time and subsequently the outcome.

It was confusing to know where to go after the injury and to trust that the communication between different units was adequate. Being referred between different units felt unsafe. Handling the practicalities around surgery like transportation and preoperative wash was a struggle and was reported to give rise to stress. The short time allowed with the physician, and the quick stay in the hospital left some feelings of not being ready to handle things by themselves. An example was the situation of discharged from the hospital with the effects of a plexus brachialis nerve block still working. The patients suffered from not yet having control of the arm and not knowing what it would be like when the effect diminished. "I couldn't get dressed when I was discharged, and it (the arm) just hanged there, and I was disgusted by my own arm. The tears gushed. It was awful" (PN 003.) To be left alone when the pain was overwhelming in the middle of the night with no one to call or no help to get was described as troublesome "I got painkillers for the first night and morning. And that night–oh my god! I had heard it is common with pain when the anesthesia wears off. But it was insane! . . . I had no idea it would be that horrible. My husband tried to call the hospital to get support." (PN 002)

The waiting time in the emergency unit was strenuous but often understood and accepted. It led to inconvenience to whom had children or pets to attend to.

Getting in touch with the surgeon afterwards led to major concerns. Even after repeated efforts to contact the hospital many participants had failed to reach the treating physician and concluded that their doctor didn't care or wished to help.

**Desire for knowledge.** Participants described an important desire for knowledge about the injury and prognosis. They also wanted to know how to get more information. How the participants perceived their prior level of knowledge impacted the expectation of the injury and recovery. All knowledge they had about this kind of injury and the recovery process was often from seeing a recovery of a relative or acquaintance who had previously suffered a distal radius fracture. For instance, participants described that they thought that they would be well immediately after surgery or cast removal and the amount of rehab and time needed for recovery came as a surprise. "When it is the first time you suffer a fracture, you believe, that when the cast is removed it would almost be over, but it was at that time the reality begun" (PN 001) The expectations could be altered by contact with the healthcare providers. When the information provided was perceived as being easy to understand and correct, it was helpful. Otherwise, it could lead to additional confusion.

Participants described a wish for a fundamental education of the injury, the recovery, and the normal process of fracture healing. "If I understood the healing process, I would feel confident that it would turn out well in the end and that is important. And not feeling like a fool hanging around waiting for something I don't know anything about. . ." (PN 011) A further understanding of prognosis and what that prognosis meant would make some participants feel more content. One example was to know whether it was harmful or not having to wait for surgery. Descriptions occurred of a wish to know what kind of pain that was normal and a wish to manage to know when to suspect that something was wrong. A consequence of not knowing about the healing process was hesitation to fulfill the rehabilitation plan in fear of making the condition worse.

Another thought was that stress and anxiety could be reduced by knowing more about how the health care system works, what to expect and who to contact with questions. Participants proposed patient information in a flowchart format, or ideally as a digital real time update regarding the referrals and wait for surgery. "If you could like. . . 'this is how the process looks like. . .' you could almost trace it digitally–'the referral has been received. You should to this and that. In case a Covid-test is required–this is how to do it' . . . Than you wouldn't have to worry for the practicalities." (PN 016)

Information provided at the time of surgery was not fully understood nor memorized, and the patient information flyer distributed was described as insufficient. There was an overall desire to get valid information both orally and written. There was also a wish to receive information digitally with links to more extensive sources for those who wanted to learn more. "...I think I would like to get it digitally; it does not have to be self-produced it could be "watch this YouTube clip". There is plenty of good material, but it is difficult to know if that is evidence based or not." (PN 011)

Some participants believed that it would be easier to learn from another patient who has recently gone through the same experience than from doctors and other health care providers. It was considered to be easier to relate to and understand each other e.g., during group rehab sessions. "It felt very good with the rehab group where we could share thoughts between each other. It was really good; we need more of that" (PN 006)

## Personal circumstances affecting the experience

The distal radius fracture does not exist on its own but is an injury that affects a person who is part of the society with different contributing circumstances influencing the experience and the recovery.

**Life circumstances.**   Having children to care for reduced the possibility to take their own time to recover which yielded a lot of pressure especially during the first painful days or weeks. Participants were affected quite some time by the stress of not being able to care for themselves as usual. "As a parent I want to take care of my children and it felt like I wasn't able to do so." (PN 012) Having a life partner or a friend nearby to support and help was a relief. During the recovery time a loss was experienced especially in individuals for whom identity in life and hobbies were built around sport activities. Having the economic resources to get appliances or take aways to facilitate life during the recovery was also described to reduce the stress induced by the lack of independency.

Having additional health issues was also described to affect the experience and expectations.

**Work related circumstances.**   Having a mentally or physically demanding work led to longer time away from normal life. Using hand operated tools in the affected hand was described to be a struggle for a long time. Even a job that did not require hand strength could be affected. Trying to stay focused or to help someone else was described as a struggle "I am a psychologist and talking to people who are suffering when I wasn't really functioning didn't work." (PN 005) To have the possibility to get back earlier and to adjust or work from home felt good and safe.

"I felt I could start to work the same week. I was working from home at a computer, and I mostly use my right hand......because I couldn't just lie there." (PN 003)

An extra stress was yielded for self-employed individuals since the inability to perform affects the company for an extended period. "It is not only to be on sick leave for two months and then go back to work. No, it's more like not being able to work for two months and then be out of clients. I had to reintroduce myself to the market and hope..." (PN 011)

**Reflection around personality attributes affecting the experience.**   The attitude towards the injury, rehabilitation and recovery could also have influenced the perceived outcome. To be diligent with the rehab and to do the exercises no matter what was described as a step forward. To be able to find new solutions in different situations reduced the feeling of being limited in different activities. The general outlook on life including thoughts primarily focused on what was lost after the injury rather than what had been achieved since the start of the rehab was also described to be important. Taking the responsibility for the outcome or placing it on others differed and could affect the experience.

## Discussion

This qualitative study showed that not only the wrist was affected when a person suffered long term impairment after a distal radius fracture, but the overall wellbeing was reported to be reduced. The impairment involved a vague but clearly noticeable feeling of disconnection or a band-like tight feeling from the wrist, both of which are difficult or even impossible to detect or quantify with traditional outcome measures. Moreover, a feeling of being marginalized, lonely or insecure negatively impacted the experience and delayed the rehabilitation after a distal radius fracture. The participants had a strong wish for more availability and information from the health care providers. Furthermore, we identified a need to confirm or clarify to patients that caregivers see the patient in the bigger picture as a person with responsibilities and activities, and not only as another radius fracture. Even if the fracture treatment was a radiological success the study participant still described limitations and a constant awareness of the wrist.

In analogy with findings exploring experience after a hip fracture, existential concerns including exploring one's mortality and feeling of fragility were present in descriptions in this study [24–26]. This finding was somewhat surprising considering that a distal radius fracture does not seem life-threatening in the perspective of the care giver. The new knowledge that this study contributes with may shed light on a need for a more holistic approach in the acute care for distal radius patients, which may have positive effects on rehab, recovery, and long-term prognosis, and in the long run a possible lowering of costs due to shortened sick-leaves and extra out-patient clinic visits. Our finding resembles those of Andreasson et al [16] in contexts such as the struggle with household chores and anxiety, but the more subtle feeling of the hand not belonging or being as reliable as before was more explicitly described by this group of patients.

Randomized controlled studies keep failing showing superiority of one treatment over another for treatment of distal radius fractures. Even in comparisons between different treatments as external fixation versus volar plating or casting versus surgery, short and long term results have shown to be surprisingly similar, even if our clinical experience depicts a completely different situation. For example, the external fixator is being described as a terribly cumbersome treatment method with a large risk of infection and malunion, especially with an axial shortening of the radius with a possible subsequent ulnar pain, whereas the volar plate seems to be very well accepted by patients with a quick postoperative recovery. Still, little scientific proof exists to support that volar plating is superior to the external fixation. The main outcomes in modern studies are Patient Reported Outcome Measures (PROMs) created in order to evaluate upper extremity problems. Many of them have been created by interviewing physicians and experts in wrist or upper extremity surgery. To the best of our knowledge there is no frequently used PROM for evaluation of wrist conditions that has been created in cooperation with patients i.e., putting the focus on what patients think is most important to evaluate after an injury. One may question if we understand our patients well enough to discern differences between treatment methods, and maybe our frequently used outcome measures have unacceptable inherent lack in face validity and relevance for evaluation of impairment after distal radius fractures. Hopefully, the present study may yield new knowledge regarding what aspects that are pertinent to include in future evaluation measurements.

### Methodical considerations

By choosing a qualitative approach the researcher is allowed to explore experiences rather than making comparisons of objective findings and draw conclusions. The deeper exploration that

qualitative investigation offers has a potential to reveal new aspects difficult or impossible to find in a quantitative testing. To what extent the findings from this qualitative study may affect future evaluation, recovery or satisfaction of patients is yet to be investigated. This is a study with purposeful sampling and the findings must be interpreted with that in mind.

To assess qualitative studies trustworthiness needs to be considered. It includes creditability, confirmability, dependability and transferability [27]. The creditability, i.e., the confidence that the reported findings were accurate, was enhanced owing to the large amount of data collected during the mean of 37 minutes' interviews. Time from injury to interview varied as well as treatment, which provided large variation and a rich data. The confirmability was increased by the researchers having different academic backgrounds and all findings were discussed in a multi-professional group until consensus was reached. Lastly, dependability was strengthened by the interview guide and the fact that the same interviewer was used throughout the study. The participants and setting are described in detail to facilitate to the reader to determine whether the results are transferable to similar contexts.

Any study has limitations that must be recognized. Our study population was difficult to reach, and though great effort was put into making it as purposeful as we suggest in this paper, we cannot be secure that we succeeded.

## Conclusion

Patients may suffer from considerable impairment difficult to detect after a distal radius fracture even if the radiological outcome is successful. During the treatment and recovery process their experience as being patients, and their personal circumstances all play important role in how the participants perceive their recovery. A perceived low level of information and knowledge might have a negative impact on recovery.

## Supporting information

**S1 Appendix. Demographics of the participants.**
(DOCX)

## Author Contributions

**Conceptualization:** Hanna Südow, Cecilia Mellstrand Navarro, Sari Ponzer, Caroline Olsson.

**Data curation:** Hanna Südow.

**Formal analysis:** Hanna Südow, Cecilia Mellstrand Navarro, Caroline Olsson.

**Funding acquisition:** Cecilia Mellstrand Navarro.

**Investigation:** Hanna Südow, Caroline Olsson.

**Methodology:** Hanna Südow, Cecilia Mellstrand Navarro, Sari Ponzer, Caroline Olsson.

**Project administration:** Hanna Südow.

**Supervision:** Cecilia Mellstrand Navarro, Sari Ponzer, Caroline Olsson.

**Visualization:** Hanna Südow.

**Writing – original draft:** Hanna Südow.

**Writing – review & editing:** Cecilia Mellstrand Navarro, Sari Ponzer, Caroline Olsson.

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
