## [Editor Report · Decision Letter 0]

28 Mar 2024

PONE-D-24-06233

Patients’ experience of suffering a distal radius fracture with long-lasting impairment

- a qualitative study

PLOS ONE

Dear Dr. Südow,

Thank you for submitting your manuscript to PLOS ONE. After careful consideration, we regret to inform you that your manuscript does not meet our criteria for publication and must therefore be rejected.

Eventhough the merit of the study is interesting, it is based on the individual patient perception, which would benefit from larger sample and a proper stastystical analysis. Patient selection criteria is missing and the mixture of surgically and conservative treated patients bias your sample.

I am sorry that we cannot be more positive on this occasion, but hope that you appreciate the reasons for this decision.

Kind regards,

Vojtech Kunc, Ph.D., M.D.

Academic Editor

PLOS ONE

---

## [Decision Letter · Decision Letter 1]

7 Aug 2024

PONE-D-24-06233R1Patients’ experience of suffering a distal radius fracture with long-lasting impairment

- a qualitative studyPLOS ONE

Dear Dr. Südow,

Thank you for submitting your manuscript to PLOS ONE. After careful consideration, we feel that it has merit but does not fully meet PLOS ONE’s publication criteria as it currently stands. Therefore, we invite you to submit a revised version of the manuscript that addresses the points raised during the review process.

There are minor changes to be addressed. Please see the comments of reviewer 1.

Please submit your revised manuscript 30 september, 2024 If you will need more time than this to complete your revisions, please reply to this message or contact the journal office at plosone@plos.org. Please include the following items when submitting your revised manuscript:A rebuttal letter that responds to each point raised by the academic editor and reviewer(s). You should upload this letter as a separate file labeled 'Response to Reviewers'.A marked-up copy of your manuscript that highlights changes made to the original version. You should upload this as a separate file labeled 'Revised Manuscript with Track Changes'.An unmarked version of your revised paper without tracked changes. You should upload this as a separate file labeled 'Manuscript'.If applicable, we recommend that you deposit your laboratory protocols in protocols.io to enhance the reproducibility of your results. Protocols.io assigns your protocol its own identifier (DOI) so that it can be cited independently in the future. For instructions see: https://journals.plos.org/plosone/s/submission-guidelines#loc-laboratory-protocols. Additionally, PLOS ONE offers an option for publishing peer-reviewed Lab Protocol articles, which describe protocols hosted on protocols.io. Read more information on sharing protocols at https://plos.org/protocols?utm_medium=editorial-email&utm_source=authorletters&utm_campaign=protocols.

We look forward to receiving your revised manuscript.

Kind regards,

Hans-Peter Simmen, M.D., Professor of Surgery

Academic Editor

PLOS ONE

Journal Requirements:

"Open access funding provided by Karolinska Institute. This study was funded by the Swedish Research Council and the Regional Agreement on Medical Training and Clinical Research between the Stockholm County Council and Karolinska Institute (ALF). CMN was supported by Region Stockholm (clinical research appointment)"

"NO authors have competing interests"

5. We note that you have indicated that there are restrictions to data sharing for this study. For studies involving human research participant data or other sensitive data, we encourage authors to share de-identified or anonymized data. However, when data cannot be publicly shared for ethical reasons, we allow authors to make their data sets available upon request. For information on unacceptable data access restrictions, please see http://journals.plos.org/plosone/s/data-availability#loc-unacceptable-data-access-restrictions.

Additional Editor Comments (if provided):

Reviewers' comments:

Reviewer's Responses to Questions

**Comments to the Author**

1. If the authors have adequately addressed your comments raised in a previous round of review and you feel that this manuscript is now acceptable for publication, you may indicate that here to bypass the “Comments to the Author” section, enter your conflict of interest statement in the “Confidential to Editor” section, and submit your "Accept" recommendation.

Reviewer #1: (No Response)

Reviewer #2: All comments have been addressed

2. Is the manuscript technically sound, and do the data support the conclusions?

Reviewer #1: Partly

Reviewer #2: Yes

3. Has the statistical analysis been performed appropriately and rigorously? 

Reviewer #1: N/A

Reviewer #2: N/A

4. Have the authors made all data underlying the findings in their manuscript fully available?

Reviewer #1: Yes

Reviewer #2: Yes

5. Is the manuscript presented in an intelligible fashion and written in standard English?

Reviewer #1: Yes

Reviewer #2: Yes

6. Review Comments to the Author

Reviewer #1: Although this is an interesting qualitative study, the included numbers are small.

It would have been benificial, if some scores would have been included, like the DASH score, SF 36 and the EQ-5D-5L score. Please add a comment in your conclusion that this is a further limitation of the study and explain why scores these scores were not performed.

Reviewer #2: 1) General comment

The authors would like to show in their qualitative study protocoll “Patients` experience of suffering a distal radius fracture with long-lasting impairment” that the measurement of a such impairment after a distal radius fracture is very difficult. And the experience as being patients, the level of knowledge and their personal circumstances play a additional role in understanding how the participants experienced their recovery.

In this qualitative study the authors show how the view of those affected is and put this in the center of the research interest. They collect the data and use it dynamically to providing the hypothesis and theories. Their data explore the directions of patients with distal radius fractures which are operated or conservatively treated. The interviews and qualitative content analysis underline their conclusions by means of their results.

The study design is very well thought out or worked out for the questions mentioned. However, it is worth discussing about the persistent impairment of distal radius fracture without pathological radiological signs, being a patient and their personal circumstances. It is very interesting how affects these facts the recovery process of the patients.

There are a lot of quantitative studies describing the correlation between radiological findings and clinical outcomes. But it does exist only a few single studies in which the personal circumstances or being a patient influence the recovery process of any fractures.

However, it is necessary to investigate these correlation by interviewing patient to explore their personal needs and experiences. And further study protocols could lead to a better understanding of these connections. Based on this conclusions there is a possibility to a personalized treatment algorhythm. And maybe it could to a more optimal outcome of such patients with a distal radius fracture with a persistent clinical impairment

Finally it would save some economical costs if these patient could get earlier to work, to reach faster their independence and to avoid additional healthcare.

2) Specific suggestions

The qualitative study mentioned above is well designed and will be appropriate for the intended research question and outcome results. Their results by interviews underline the conclusion that patients suffering from considerable impairment could be difficult detected after a distal radius fracture even if the radiological outcome is successful. The three main categories are specifically worked out by the authors. However, persistent impairment and how it affects life, their experience as being patients and their personal circumstances play an important role in how the participants perceice their recovery.

7. PLOS authors have the option to publish the peer review history of their article (what does this mean?). If published, this will include your full peer review and any attached files.

Reviewer #1: No

Reviewer #2: No

---

## [Author Response · Author response to Decision Letter 1]

11 Sep 2024

This is a good point that PROMs and other additional quantitative data is missing in this qualitative study. 

When we started the study, we collected both DASH, PRWE, EQ-5D HAD. Any proper statistic, in this setting with the purposeful sampling, would however per definition be biased. We will provide the PROM and radiograph as demographics as an appendix for anyone to learn more about the material, line 162.

The data cannot be be published due to restriction in the ethical approval. This kind of data cannot be properly anonymized due to the format of transcribed interviews. We added an adress to request the data any request will be submitted in an request to the Swedish Ethical Review Authority for permission to share data

---

## [Editor Report · Decision Letter 2]

18 Sep 2024

Patients’ experience of suffering a distal radius fracture with long-lasting impairment

- a qualitative study

PONE-D-24-06233R2

Dear Dr. Südow,

We’re pleased to inform you that your manuscript after revision 2 has been judged scientifically suitable for publication and will be formally accepted for publication once it meets all outstanding technical requirements.

Kind regards,

Hans-Peter Simmen, M.D., Professor of Surgery

Academic Editor

PLOS ONE
---

## [Editor Report · Acceptance letter]

23 Sep 2024

PONE-D-24-06233R2 

PLOS ONE

Dear Dr. Südow, 

I'm pleased to inform you that your manuscript has been deemed suitable for publication in PLOS ONE. Congratulations! Your manuscript is now being handed over to our production team.

Kind regards, 

on behalf of

Dr. Hans-Peter Simmen 

Academic Editor

PLOS ONE